# Development of an Artificial Neural Network for the Detection of Supporting Hindlimb Lameness: A Pilot Study in Working Dogs

**DOI:** 10.3390/ani12141755

**Published:** 2022-07-08

**Authors:** Pedro Figueirinhas, Adrián Sanchez, Oliver Rodríguez, José Manuel Vilar, José Rodríguez-Altónaga, José Manuel Gonzalo-Orden, Alexis Quesada

**Affiliations:** 1Departamento de Patología Animal, Universidad de Las Palmas de Gran Canaria, Trasmontaña S/N, 35416 Arucas, Spain; pedro.figueirinhas@fpct.ulpgc.es (P.F.); oliver.rodriguez@ulpgc.es (O.R.); 2Department of Computer Science and Institute for Cybernetics, Campus de Tafira, Universidad de Las Palmas de Gran Canaria, 35017 Las Palmas, Spain; adriangsanchez@gmail.com (A.S.); alexis.quesada@ulpgc.es (A.Q.); 3Departamento de Medicina y Cirugia Veterinaria, Campus de Vegazana, Universidad de León, 24071 León, Spain; jarodma@unileon.es (J.R.-A.); jmgono@unileon.es (J.M.G.-O.)

**Keywords:** artificial neural network, web app, lameness, dog, inertial sensor

## Abstract

**Simple Summary:**

Many of the rules considered valid in the ambit of lameness detection in domestic animals are mainly subjective or acquired after extended clinical experience. Thus, it is necessary to develop tools and/or technologies to provide objective data to discern between sound and lame animals. The main objective of this study is to develop an artificial neural network by analyzing data obtained from an inertial sensor in order to discriminate between sound and lame dogs. After different system adjustments, the neural network has been able to correctly determine whether dogs were lame in about 86% of cases. Furthermore, a web app was developed to manage and follow the different cases. An artificial neural network was designed and developed to analyze spatial data from inertial sensors and to detect motion alterations in dogs with unilateral lameness. Displayed within a user-friendly, intuitive web app, the system could be a useful tool for lameness detection for veterinary clinicians.

**Abstract:**

Subjective lameness assessment has been a controversial subject given the lack of agreement between observers; this has prompted the development of kinetic and kinematic devices in order to obtain an objective evaluation of locomotor system in dogs. After proper training, neural networks are potentially capable of making a non-human diagnosis of canine lameness. The purpose of this study was to investigate whether artificial neural networks could be used to determine canine hindlimb lameness by computational means only. The outcome of this study could potentially assess the efficacy of certain treatments against diseases that cause lameness. With this aim, input data were obtained from an inertial sensor positioned on the rump. Data from dogs with unilateral hindlimb lameness and sound dogs were used to obtain differences between both groups at walk. The artificial neural network, after necessary adjustments, was integrated into a web management tool, and the preliminary results discriminating between lame and sound dogs are promising. The analysis of spatial data with artificial neural networks was summarized and developed into a web app that has proven to be a useful tool to discriminate between sound and lame dogs. Additionally, this environment allows veterinary clinicians to adequately follow the treatment of lame canine patients.

## 1. Introduction

Visual evaluation of lameness varies greatly among observers, which fails to agree with objective measurements of limb function, such as kinetics and kinematics. However, there is an increasing interest in better understanding canine lameness by using more precise and objective methods of assessment [1,2].

A recent kinematic study in dogs with experimentally induced lameness performed with a motion capture system demonstrated that symmetry measurements of vertical head and pelvic movement, which are clinical variables commonly used in the visual assessment of lameness, were enough to identify the lame limb [3].

Inertial measurement unit, or inertial sensor, (IMU) systems have been developed commercially to evaluate objective lameness in horses [4]; IMU systems are based on sensor technology and comprise gyroscopes, accelerometers, and magnetometers [5,6,7]. The commercially available IMU for horses has been experimentally used in dogs [8]. Other IMU systems have also been used to detect lameness in sheep by measuring the general activity of these animals [9].

The use of artificial intelligence in the medical field has enabled the development of different systems of medical support for early diagnoses of skin tumors and cardiovascular diseases [10,11,12]. However, there is no commercially available software for lameness detection in dogs in the same manner as in horses.

Theoretically, artificial neuronal networks try to emulate human neuronal behavior. The internal “structure” has some similarities [13]: the dendrites in human neurons receive signals from other neurons. These signals are processed in the cell body, and a signal is finally propagated by means of axonal connections to dendrites of other neurons (Figure 1).

Likewise, in an artificial neuron, data are received as input; then, mathematic algorithms are applied in the artificial neuron “body”, producing an output. These algorithms are based on linear regression [14]. Conversely, the *bias* is an added constant that aims to adjust the neural output. This addition tries to improve the result in order to mimic a real situation [15].

Instead of utilizing a single neuron, multiple artificial neurons are grouped together in an artificial neural network (ANN), which is organized in layers [16]; for example, the *input layer* receives input data and provides the parameters for analysis, the *hidden layer* processes input data, and the *output layer* provides the output results classified in categories (Figure 2).

In order to correctly predict different categories with accuracy, it is necessary to “train” the network. For this purpose, it is vital to have a large amount of varied training data [17]. In this way, the network will progressively “learn” from these input data. With the *loss function*, network output data are compared with expected output.

Currently, to the authors’ best knowledge, ANN technology has not been developed for lameness detection in dogs.

Based on these premises, the main purpose of this study is to develop an ANN, which with spatial data obtained from IMUs, can discern whether a dog has unilateral hindlimb lameness. Second, this study seeks to develop a web portal with the capacity to manage dog lameness data with the aim to follow up with the dog over time.

The study presented here has been designed as a “proof of concept”, whose initial development has required multiple adjustments in many parameters until satisfactory performance is obtained; for this reason, the sections of materials, methods and results are merged.

## 2. Materials, Methods and Results

### 2.1. Instrumentation

For data recording, an IMU (MTw Awinda Wireless 3DOF Motion Tracker, Xsens**^®^**, Enschede, The Netherlands) was used (Figure 3).

All data were collected by a dedicated software (MT Manager**^®^**, Xsens**^®^**, Enschede, The Netherlands). Data were exported with the extension. *txt.*

The IMU was positioned above the midline of the spinous processes of the sacral vertebra 2 with adhesive tape. Sensor data were digitally sampled (8 bit) at 100 Hz in real time. The device has dimensions of 4.7 × 3 × 1.3 cm and a mass of 16 g.

To observe data patterns obtained with the sensor in order to discriminate which relevant elements distinguish sound and unilaterally hindlimb lame dogs, the movement amplitude of the rump from both sides was measured (Figure 4).

The gyroscope provides orientation data in a longitudinal direction of *pitch*, *yaw*, and *roll*. From these, the *roll* provides angular data of the lateral inclination (Figure 5).

### 2.2. Dogs

Fifteen working dogs weighing more than 15 kg were recruited for this study. Dogs weighing less than 15 kg were excluded from the study, since these animals feel the presence of the sensors more when moving, creating a “perturbed” locomotion. From the 15 working dogs, 12 dogs were used for training the ANN—7 of them were sound and 5 were lame; 3 additional dogs were used for the verification phase. Lame dog owners were clients of the Veterinary Teaching Hospital of the Universidad de Las Palmas de Gran Canaria (Spain).

### 2.3. Procedure

The dogs were walked straight on a leash for enough time to reach a uniform gait, next to the handler, without pulling on the leash and with the head straight; to put the dogs at the “same dynamic status”; velocity was progressively augmented to reach a maximum where the dogs were still at walk. During a walk, the alternate movements of both hindlimbs create a successive inclination on the sensor toward both sides, generating the inclination data (*roll*). Below, the inclination data for multiple consecutive steps is visualized graphically (Figure 6).

As shown in Figure 6, the ascending and descending patterns show a certain consistency, given that the maximum and minimum inclination values are similar. However, certain isolated values appear to be out of this pattern; these anomalies can be attributed to head movements due to distractions, head shakings, etc. Data shown as boxplot graphics have a range of about 20 degrees of inclination. However, a substantial number of outliers can be observed; this fact could generate a bias when the system attempts to generalize input data. For this reason, we pre-processed the test data to eliminate “noise” or outlying values. Regarding the outlier criteria, the process carried out the use of the median of the sequence, and a cutoff was made on the percentiles, taking into account that the sampling to be used had values between 10% and 90% of the sampling. (Figure 7a,b).

With this procedure, extraneous data not attributable to necessary motion were discarded (Figure 8).

#### 2.3.1. Neural Network

##### Design

Data obtained with the IMU were a temporal series, in which certain data depended on previous data; for this reason, a recurrent neural network (RNN) was used because this type of ANN is able to “remember” previous events [18,19]. Specifically, an RNN variant known as Long Short-Term Memory (LSTM) was chosen because it offered better performance over other RNN variants [20].

The LSTM parameters initially set were the following:Number of inputs: 1. The sequence of consecutive data was set to 250 as a single input.LSTM layers: 2.*Dropout*: 50%. This technique randomly “freezes” a determined number of neurons (in our case 50%) so that these neurons are ignored during the updating parameters phase. This produces neurons that are still working and are more relevant, highlighting slight changes in the training data and improving the network’s ability to generalize data. As mentioned before, this technique was used exclusively during the training phase [20].Number of outputs: 2. Since the network output may be any numerical value, it has been normalized using a sigmoid function, giving a 0 value for sound dogs and a 1 value for lame dogs. For the final diagnosis of sound or lame, the LSTM network is complemented with a traditional ANN, known as the Fully Connected Layer (FCL).Number of neurons in each cell: 256. As is usual in many artificial neural networks, this parameter was obtained empirically. We chose a starting power value of 2, and through an intensive battery of tests, we chose the value that offered the best results while maintaining a reasonable computational cost.

In relation to the neural network, there is a single input at each iteration along the LSTM network, the value of which represents the degree of rump slope at each point in time. Thus, the inputs correspond to the degrees of the hindlimbs at each moment (23*, 35*, etc.) while the outputs by the network are numerical values that in our case we normalized to values between 0 and 1, thus obtaining as output a value that represents the probability that the dog has a lameness or not.

Concerning the number of layers of the LSTM network, two were used. Using a lower value (1) could cause the network not to sufficiently learn from the training data due to the low complexity of the network, and this would cause it to not be able to generalize the training data correctly. Conversely, this value was not increased for two reasons: to avoid excessive computational cost in the training phase due to the number of neurons and to avoid overtraining, which could cause the network to memorize the training data, thus impairing the generalization of the system.

Most of the training data had more than 400 data obtained from the sensors, each datum refers to the degree of inclination of the sensor at each moment of the sequence, but it was decided to limit this sequence to 250 data in order to eliminate the data from the beginning of the acquisition, since these could have certain “anomalies” due to abrupt movements of the dog at the beginning. If any sequence had less than 250, then it was filled with null values (0) until the 250 were reached. We used supervised learning where we used exactly 250 data obtained from the sensors to train the network. If more data were available, they were used to generate more training data.

The implementation of the neural network using the library *Pytorch* in *Python is showed* (Appendix A).

##### Training

Prior to the training phase, *cost* and *optimization* functions had been used. The *cost function* evaluates the performance of the model. It takes both predicted outputs by the model and actual outputs and calculates the extent of error in the model’s prediction. Essentially, it is a measure of the network’s success with the training set, where a value near 0 means that the network makes accurate predictions during training. For this purpose, the mean squared error formula was used, which is the squared sum of the difference between the real result and the prediction generated by the neural network, i.e. $MSE=\frac{1}{n}\overset{n}{\underset{i=1}{\sum }}{\left(y_{i}-{\tilde{y}}_{i}\right)}^{2}$.

Once the error has been obtained, the *weights* of the neural network should be adjusted; the *weights* determine the relevance of each network input in order to obtain better predictions. For this purpose, it is necessary to select a function that updates the weights. This is called an *optimization* function. *The gradient descent* algorithm is mostly used for these purposes [21]. In our study, we specifically used the *Adam algorithm*, which is one of the most widely used optimization algorithms due to advantages such as high efficiency and low resource consumption [22].

After this, the real and generated output were compared, and a value was given for the difference. This difference value is called *loss* and serves to adjust the *weights*. The frequency by which the weights are updated is determined by the *learning rate*, which is a *hyperparameter.* This rate was determined empirically, because setting a low learning rate makes the network learn slowly. However, a high learning rate makes the *weights* and *error function* diverge; thus, there is no learning at all. In our study, we used a standard value (0.1).

An additional hyperparameter to be settled for the neural network was the number of *epochs*, or the number of times the network receives input data in order to progressively generalize and learn from them. It was necessary to be careful setting this hyperparameter because a wrong value could lead to bad training of the neural network. In summary, *weight*, *epochs*, and *dropout* were modified in an attempt to optimize the training (Appendix A).

Thus, the first step is to set the network in training mode and allow its *weights* to be updated. After this, the training data are obtained in batches; these training data were used as input in the network to observe the generated predictions and then to check those predictions against the actual result.

During this phase, many mathematical operations are used in the network to generate output. Once this phase is finished, the *weights* of the network are updated using the *optimizer*. The aim is to adjust the weights to align with the input data; thus, the network can make better predictions in the following *epochs.*

##### Verification with Blind Data and Improvements

In this phase, it was necessary to validate the results with a validation data set different from the training data set, in order to assess the network’s ability to generalize its behavior in the presence of unknown data; in our case, the validation set represented 20% of the available data set (3 dogs). When the system was evaluated with this set, the loss was much higher than the training phase (0.368); unfortunately, the accuracy was relatively low (0.6 or 60%) (Appendix A).

For this reason, certain improvements were necessary. For instance, the technique of the *clipping gradient* was used to avoid erroneous updates of the *weights* [23]. Furthermore, more dropout was applied (60%), and the *batch size* was readjusted to three trials. With these improvements, the accuracy of the neural network improved to 85.7%. (Appendix A).

#### 2.3.2. Web Application

The diagnostic tool implemented by the ANN was integrated into a web management tool to manage and record all the information from the dogs in this study. This system allows veterinary professionals to collect relevant information about the dogs, including the videos of these animals with different recordings with IMUs to capture inclination data.

This web application used many programming languages and libraries in its development. Regarding the development of the server, the language “Python”, along with a library called “Flask”, was used. “Angular”, an open code component-based framework, was used for the user interface.

##### Elements and User Roles

Two key elements were defined for the web application development:− The *sessions*, which follow the dog’s lameness− The *trials*, three from each dog, with the IMU fixed to each dog’s lower spine. Sensory data were synchronized with video recordings for a simultaneous visualization of the graphics and the dynamic of each dog.

Two user roles were established in the application: the *administrator*, who has full access to web functionalities and has exclusive authority to register other users in the system, and the *veterinarian*, who inputs and manages data (canine registrations in the system, data updating, view the sessions of the dogs) and can create and edit new sessions, visualize the trials to observe the angular characteristics of the analyzed dogs, and observe conclusions by the neural network.

##### Architecture

The web application follows a general client–server architecture, allowing the server to manipulate data in the application and offering a wide selection of “services” the client can use to obtain, modify, delete and create data in the application. The protocol “HTTP” was used for this communication (Figure 9).

##### Database Design

For the preservation of data, the database type SQLITE3 was used. Our database has different entities. First, there is the *dog entity* that represents the dogs created in the portal. It contains descriptive data about the dog, such as its name, date of birth, breed, weight or height, as this data may be relevant when diagnosing lameness. Second, dogs have multiple *sessions* associated with them, which shows the working sessions between a veterinarian and a dog to store multiple trials of data. The trials are in charge of storing the data obtained by the sensors, as well as their subsequent processing, to show the different inclination characteristics, to process the data in the ANN, and to offer diagnoses about the presence of lameness in the evaluated dogs.

##### Data Storage

The web portal was designed to allow for file uploads, video recordings, and sensory data. With this purpose, a hierarchical structure was utilized to store the different types of information (Figure 10).

Each dog has its own folder. The name of the dog’s folder is composed of the prefix “dog folder” with the identifier of the dog added to ensure no two folders have the same name. For each session created, a folder is created with a name that follows the same structure, but in this case, the prefix is “session”. At the next level, the trials follow the same structure as above; finally, the files relating to sensors and videos also follow a similar naming pattern.

##### User Interface

Different *mockups* were developed using the tool *Figma*. The development of the user interface had all the necessary elements: a good distribution of elements, a friendly color palette, and other crucial elements for the final user, the veterinarian.

The main application view shows dogs registered in the system. The interface allows for setting different filters restricting the displayed canines (Figure 11).

When accessing a specific dog, another view containing the essential animal registered data is displayed (Figure 12). From here, a user can access the different sessions that have been recorded for that dog (Figure 13). At the same time, for each session, the user has access to the different recorded trials, where the visualization of sensor data is synchronized with the trial video recording (Appendix A) (Figure 14).

##### Server

The development of an interface with representational state transfer (*API REST*) for the server allowed for the separation of server logic from the user interface, improving the maintenance of the application. In this way, there is a project with a series of communication functionalities within the database (creation, update, deletion, and view), and there is a separate web interface where different views can be observed.

For programming, *Python* language was used, given that it allows for the installation of several specific libraries, such as *Keras*, *Numpy* or *Pytorch*. For the web application development, the minimalist framework *Flask* was used.

##### Client

For the web interface development, *Angular* was used because of its quality and well-structured software, in addition to having many facilities for its development. Because the web interface is a single page application (SPA) framework, keeping all content on a single page, the page does not require reloading the browser when accessing other routes, offering greater fluidity to the user. Regarding the libraries used for the project, “Chart.js”, a library that generates multiple graphs, was used to display IMU values. For user authentication on the web page, “Angular JWT” was used. This library makes use of a token to allow for subsequent operations requested by users once they are initially authenticated by a username/password. The tokens are verified on each request made to the server to confirm that request is authorized. Finally, “SweetAlert2” was used to create pop-up windows.

## 3. Discussion

Our study assessed the ability of an ANN to discern if a dog is sound or lame, imitating the knowledge of an experienced veterinary clinician.

Within the context of this pilot study, we chose canines supporting hindlimb lameness because the fixation of the inertial sensor in the midline of the spinous processes of the sacral vertebra 2 was easier than on the top of the head. A recent paper was able to obtain reliable data from sensors affixed to the top of the heads of dogs with induced lameness [8]; this study used a commercially available sensor-based system to detect lameness in horses [5], in which the fixation method is undoubtedly better developed than in our experimental study. If we were able to achieve an optimal fixation of the device, we believe that our system would also be able to detect forelimb lameness, but in this case, the lameness would be evidenced by the asymmetry of the pitch angular data.

Regardless of placement, the ANN should be able to analyze data from sound dogs and differentiate them from dogs with altered locomotion due to lameness. These compensatory lameness mechanisms have been described in dogs, demonstrating that compensation occurs by changing ground reaction forces of ipsilateral and contralateral limbs in an attempt to alleviate pain [24]. These changes unload the painful limb, and they can only be achieved by dynamic postural adaptations of the head, trunk and limbs, with subsequent changes in their motion patterns [3]. Specifically, our sensors detected variations (asymmetry) of the angle of the pelvis that occur in unilateral hindlimbs lameness when a dog is at walk. For bilateral lameness detection, we think that, analogous to inertial sensor-based devices used in horses [5], at least the combined data from two or more sensors would be needed, but this still requires further investigation.

This system could also be potentially used in other four legged domestic species such as cats, since the dynamics of movement are similar. The only limitation could be that the animal will alter the movement if the device causes any discomfort.

Regarding the learning ability of the neural network and management of input data, a recent study in horses [25] obtained tridimensional data from markers recorded with cameras. The study divided the input data into one section, containing 80% of all data for training and another section with the remaining 20% excluded from the training phase. The 20% section was used only for testing to give an unbiased evaluation of the network’s performance. Our study design was consistent with the previous study’s proportions.

In order to obtain reliable data, the decision was made [26] to estimate input data sets > 6000 to guarantee statistical validity. The enrollment requirements for valid dogs in our study (working dog, medium to big conformation, able to walk on a leash, etc.) made this target unattainable. To avoid the need to attain this enormous amount of data, a recent study proposed identifying and deleting data that only marginally contribute to the outcome [25]. Therefore, in our case, we decided to exclude outliers that added “noise” to the data set and made the system difficult to distinguish accurately between sound and lame dogs. Other authors [27] have previously made the same corrections in similar studies involving lameness assessment.

Conversely, some authors opine that excess data are counterproductive for system performance. A network such as ours (RNN) suffers from *Vanishing Gradient Problem* [28], where a long input data set with neurons located farther away from the output data can hardly learn something new and is unable to “generalize” the patterns of the training data. For this reason, the input data to the network should be limited during the training phase [19].

For all the reasons described above, the data sets were limited to a maximum of 250 input data from the sensor. This was accomplished by initially collecting about 400 data. After deleting the outliers as described in the Materials and Methods section, there remained approximately 250 input data. If it was necessary to reach an amount of 250 data; the empty spaces were filled with 0 starting from the 0 position, as if the dog were initially still at the start of the recording of data. In this way, these additional data did not influence the learning process.

When evaluating the ANN final performance, some authors reported a correct classification lame/sound of 78.6% in horses [25]; in our study, we reached an accuracy of 85.7% in unilaterally hindlimb lame dogs, which is a similar performance.

However, our system showed different limitations and potential improvements.

First, the data collection and treatment could be automatized; second, the final diagnostic (lame/sound) should be given instantly; and third, the lameness could be quantified as in other equine models [29].

To be sure, although the performance of the proposed system is high, the authors believe that the data are somewhat limited because they were obtained from a group of dogs with a narrow conformational range. In our opinion, the decision to utilize this system as a standard tool is premature and susceptible to commercial exploitation.

## 4. Conclusions

The results from this study show for the first time how an ANN, analyzing inclination data from an IMU affixed on the sacral vertebra 2 of a dog, is able to differentiate among sound and unilaterally hindlimb lame dogs, detecting the asymmetries in the vertical pelvic amplitude (angle) when the animal is at walk. These results could help to further develop a fully automated system for lameness detection in the future, being able to detect frontlimb and bilateral lameness also. This will help improve objective discernment in lameness assessment in dogs.

## Figures and Tables

**Figure 1 animals-12-01755-f001:**
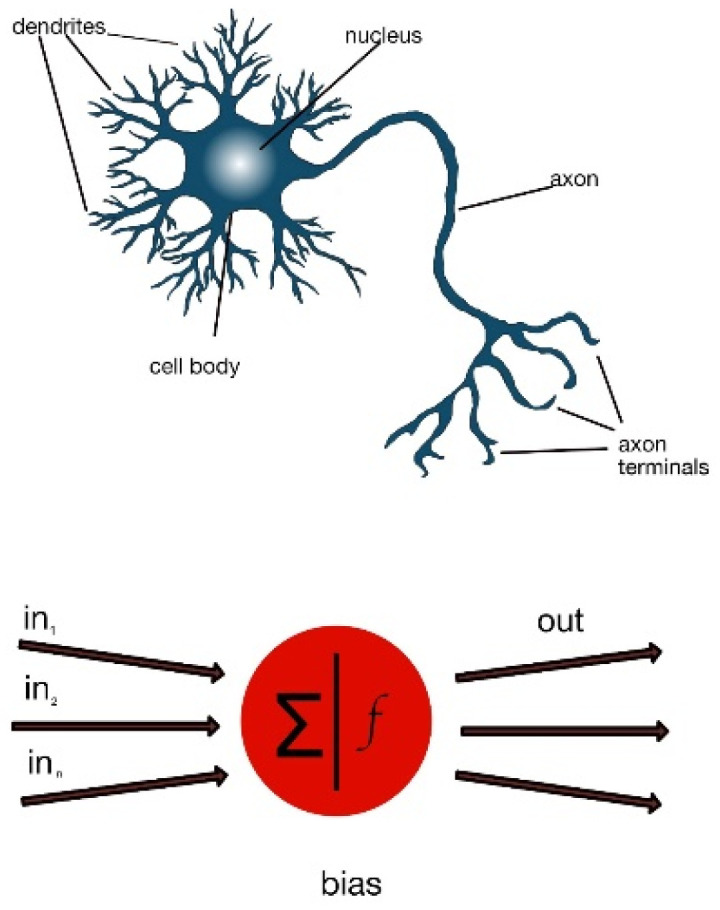
Structure of a neuron (**top**) and an artificial neuron (**bottom**).

**Figure 2 animals-12-01755-f002:**
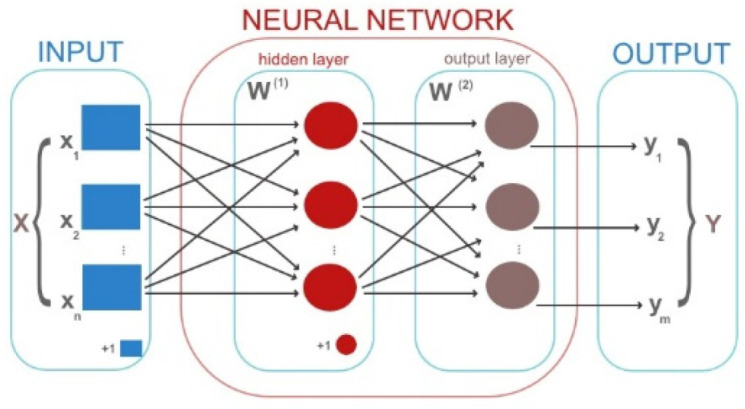
The structure of an ANN.

**Figure 3 animals-12-01755-f003:**
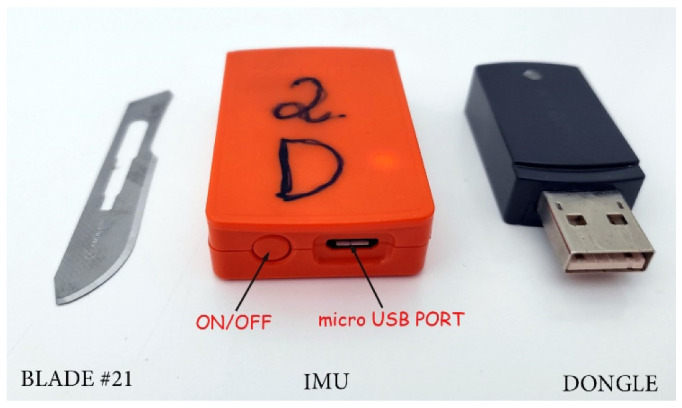
The external appearance of an IMU and the wireless connection dongle.

**Figure 4 animals-12-01755-f004:**
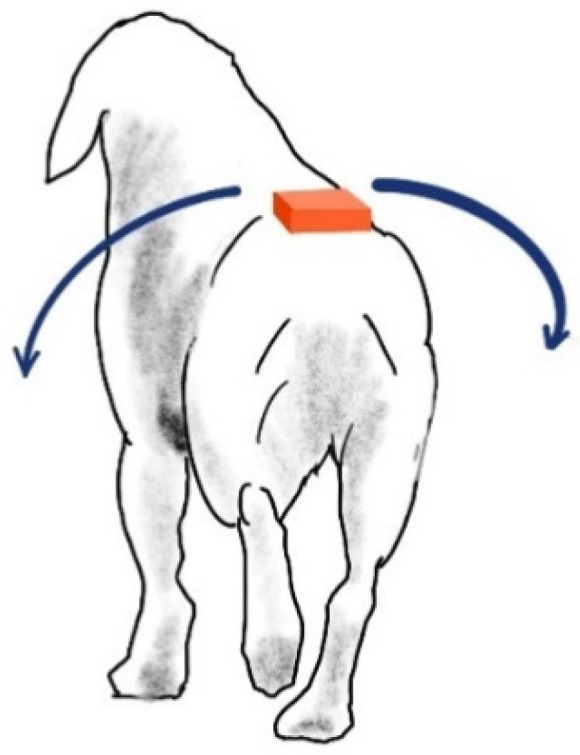
The IMU location. The bilateral inclination is measured with the dog at walk.

**Figure 5 animals-12-01755-f005:**
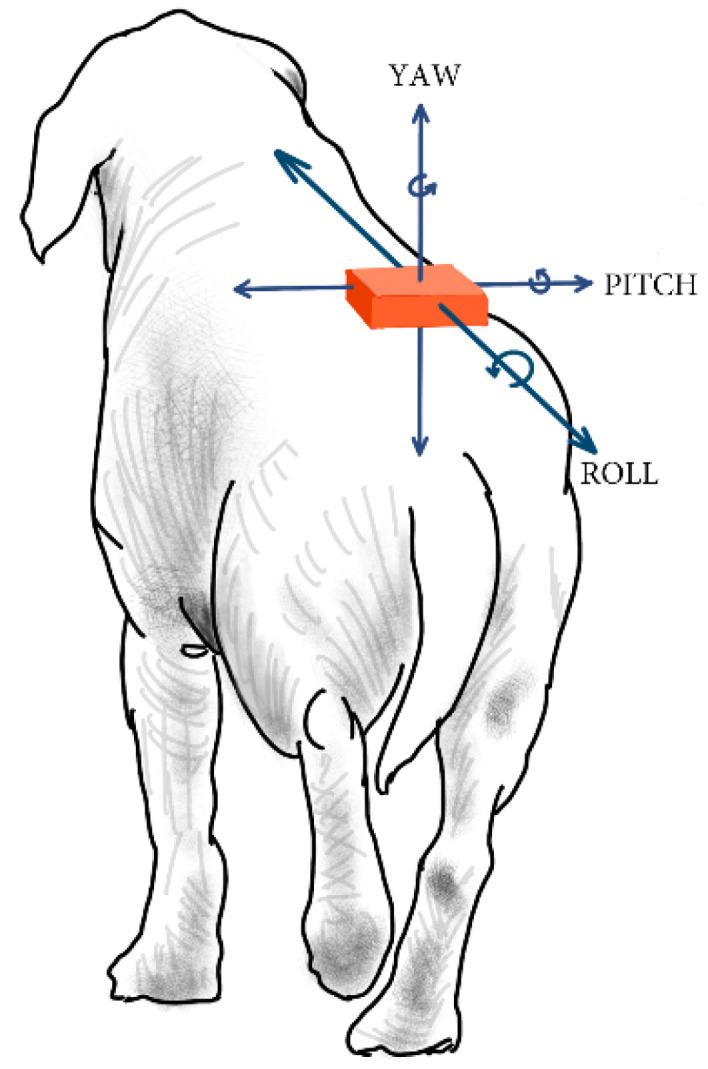
The IMU gyroscope provides inclination data on the X, Y, and Z axes.

**Figure 6 animals-12-01755-f006:**
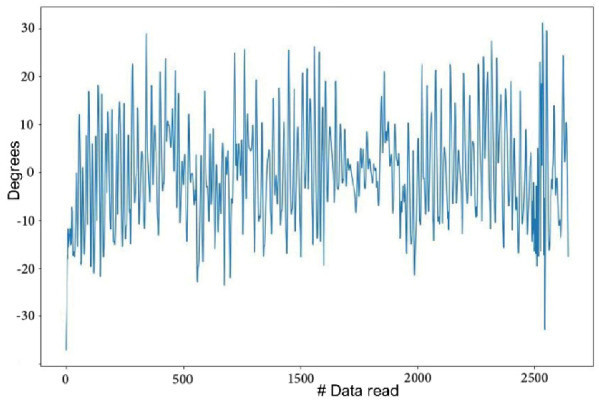
Lateral rump inclination data from multiple steps of a sound dog at walk.

**Figure 7 animals-12-01755-f007:**
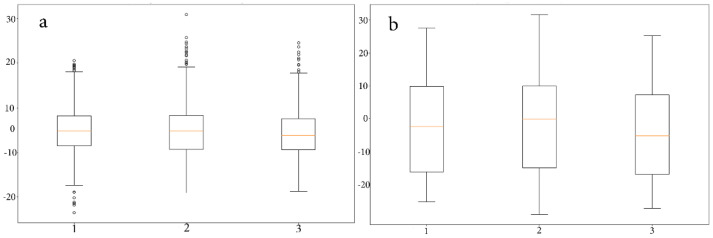
(**a**) Boxplot graphic before data cleaning. (**b**) Boxplot graphic after data cleaning, eliminating outlier values.

**Figure 8 animals-12-01755-f008:**
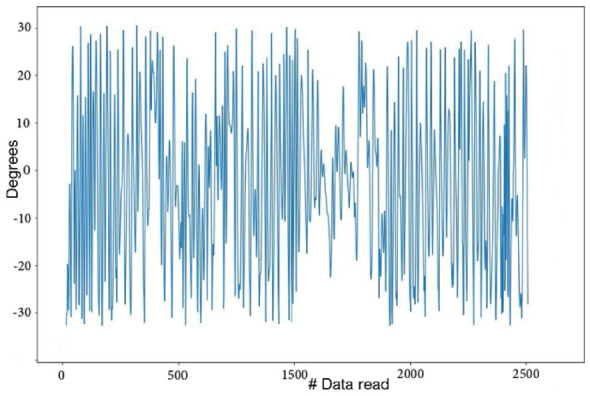
Graphic of the same sound dog after data cleaning; data symmetry has been increased compared with the data from Figure 6.

**Figure 9 animals-12-01755-f009:**
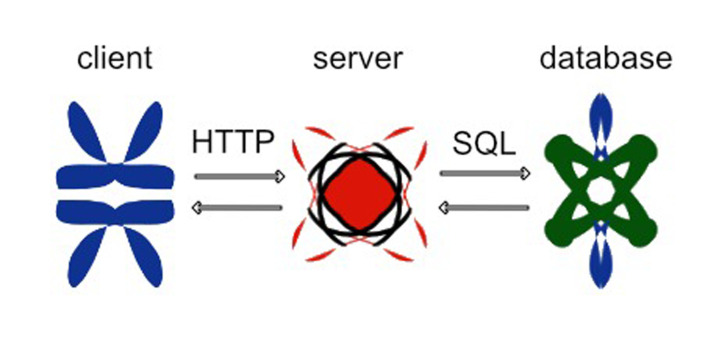
The communication scheme among the different app components.

**Figure 10 animals-12-01755-f010:**
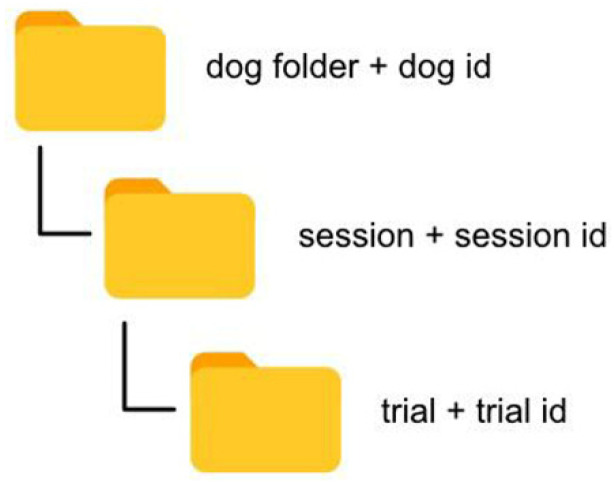
The file structure within the app.

**Figure 11 animals-12-01755-f011:**
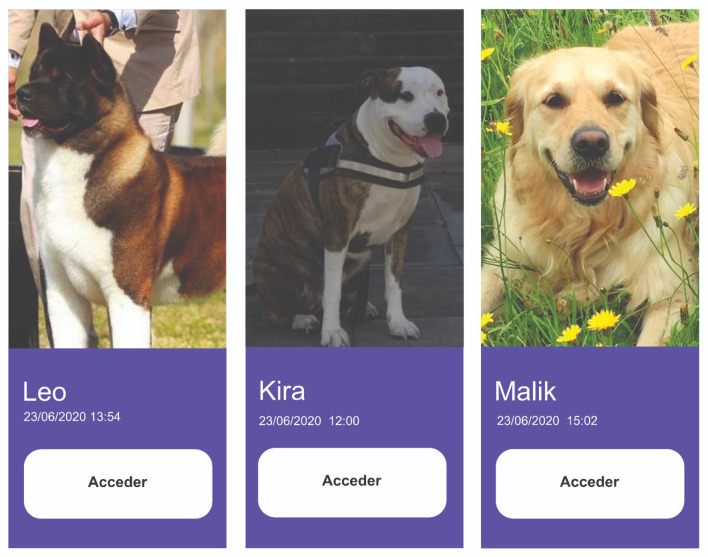
Access to different dogs registered in the system.

**Figure 12 animals-12-01755-f012:**
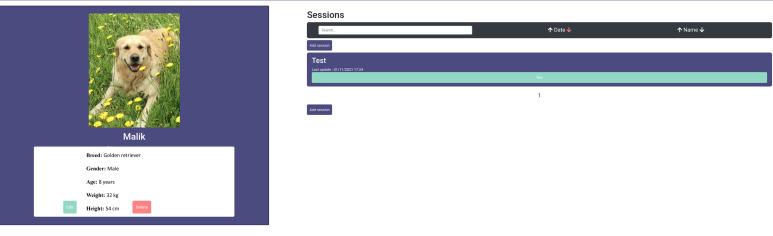
The essential data of a specific dog.

**Figure 13 animals-12-01755-f013:**
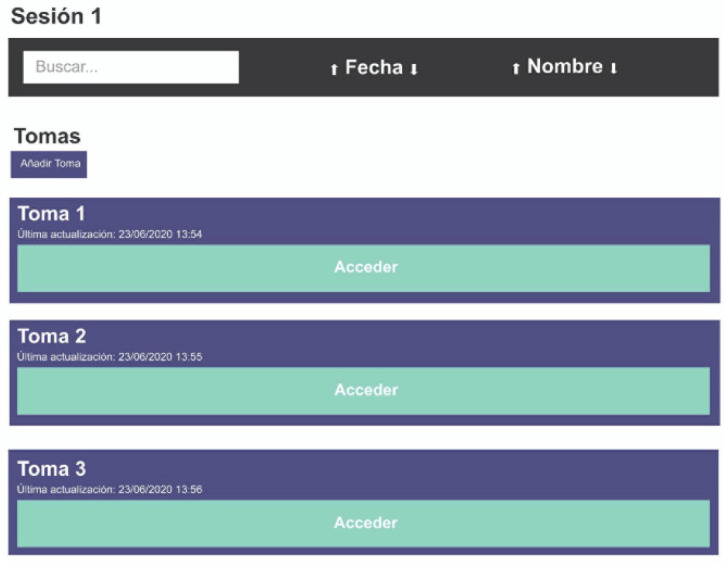
The access to sessions and trials of a specific dog.

**Figure 14 animals-12-01755-f014:**
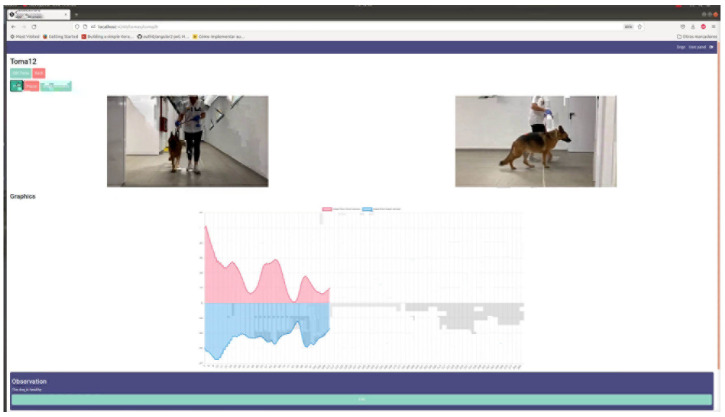
A screenshot of a synchronized graphic from IMU data and video recording.

## Data Availability

Data sets are available for researchers under reasonable request. Source code is accessible from: https://github.com/AgSanches/TFG-Backend(server); https://github.com/AgSanches/TFG-Frontend(client).

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
