# Peer review of "Development of an Artificial Neural Network for the Detection of Supporting Hindlimb Lameness: A Pilot Study in Working Dogs"

_animals, 2022, doi:10.3390/ani12141755_

Round 1

Reviewer 1 Report

The question of this study relates to the development of an artificial neural network by analysing data obtained from an inertial sensor to distinguish between healthy and lame dogs. It was investigated whether it is possible to distinguish between dogs lame at a hind limb and healthy dogs using this newly developed system.

A lot has been published regarding the lameness locator in equine medicine, but to my knowledge this is a newly invented system in dogs and could be of great benefit to small animal lameness evaluation.

I would appreciate to know more detailed about how the dogs were measured. I would like to know more details about the exact measurement process. How exactly were the dogs measured? Were they kept on a leash or were they allowed to roam freely. Was attention paid to the head position during the measurement? Could this have an influence on the force distribution of the hind legs (yes, please mention publications) - if care was taken that the dogs kept their heads straight, please mention this.

some figures (9 or 11 for example) could be of better quality but I assume it is due to the output of the App.

Reviewer 2 Report

In their manuscript, Figueirinhas et al. introduce an artificial neural network (ANN) for detecting lameness in working dogs, with the aim of enabling veterinarians or veterinarian technicians to better distinguish if an animal (dog) is lame or sound. The authors developed an ANN by analyzing data from an inertial motion sensor unit (IMU), which was able to correctly identify dogs as lame or not in 86% of cases. The authors went on to develop a web app using the ANN to analyze spatial data from IMUs to detect motion alterations with unilateral lameness, with the hope that it can be deployed as a user-friendly tool to assist in accurately detecting lameness in dogs.

The paper describes what should be an accessible and useful tool to assist veterinarians and technicians in diagnosing hindlimb lameness in dogs. Rather than being a polished final product, it seems more of a proof of concept which could be further developed before being deployed commercially to veterinary clinics. The paper should be of interest to readers and seems a good fit for a special issue on “Sports Medicine and Animal Rehabilitation”.

As general advice, the authors should have a native / fluent English speaker proofread the manuscript, particularly the first half, to correct instances with somewhat vague or incorrect wording. I listed several instances below with suggested changes in italics, but this became a lengthy proofreading task rather than a scientific evaluation, so I suggest the authors or editors enlist copyediting assistance so the language can be clarified.

The concerns raised below are all minor, but would improve the readability and bring some more clarity to the methods:

·         Several 1- and 2-sentence paragraphs throughout the manuscript should be collapsed into longer ones- this would help the flow and readability.

·         Line 35 “…. are very promising… “ should be moved to the end of the sentence. 

·         Line 50 “… commonly used in the visual assessment of lameness….”

·         Line 55 “…The commercially available IMU for horses…”

·         Line 58 “…has enabled the development of different systems of medical support…”

·         The figures in later version of the manuscript should be higher resolution and centered. In this version, nearly every figure in the manuscript was off to the left or right of the panel.

·         Line 71: should it say linear regression (with an “r”) instead of lineal (with an “L”)?

·         Line 82: could be more clearly stated that “… it is vital to have a large amount of varied training data…”

·         Figure 3:  it would be helpful to include a scale metric and some indicators of what the button(s) and slots on the IMU are for.

·          Line 98: … “the IMU was positioned above the midline of the spinous processes…”

·         Figure 5: the text for “pitch”, “roll” and “yaw” should be larger & more easily readable. It would also be helpful if these rotational axes were drawn with respect to a dog, so it’s totally clear what pitch, roll and yaw mean in terms of the animals’ own movement.

·         Line 120: “… the generated inclination data for multiple consecutive steps…”

·         Line 124: “… as shown in Figure 6, the ascending ….”

·         The text should state how the “outlier” criteria were arrived at—under 10% and over 90% of the range of observed data seems reasonable if it improves performance of the network, but it is not clear what is meant by < 10% or > 90% of “set values”.  Was this in relation to the mean or median?  How many standard deviations away from the mean were the cutoffs?

·         Figures 6 and 7:  the axis labels should be centered on each axis of the graph, not at the ends. Also: is the time axis really milliseconds, or is it seconds?

·         Line 144: “… type of ANN is able to “remember”….”

·         Line 160: The authors should explain why 256 neurons (nodes?) were used in each cell (LSTM network?).

·         Line 193: There was an extra “a” in:  “…wrong value could lead to A bad training…”

·         The training of the LSTM network was explained in a very accessible way for a general readership—explaining, for example, how loss was minimized through a process of gradient descent. However—some readers may wish to know what the “inputs” and “outputs” of the neural network correspond to data-wise.  Line 148 states that the sequence of consecutive data was set to 250 as a single input—does that mean 250 values (sampled at 100 / second) for each of the rotational axes measured by the IMU?  And how was the output evaluated as correct or not—was the learning process supervised?

·         Again, in the Discussion there were mentions of “… 250 input data”, or “…collecting about 400 data.”  Do these mean data points?  Or did each input datum correspond to 250 sampled points from the IMU? (this is essentially the same question raised just above & needs to be defined somewhere).

·         Points of general interest that could be raised in the discussion include: What about bi-lateral lameness?  Or lameness in the forelimb vs. hindlimb?  Would this work for cats or other domestic, 4-legged house pets?  

Reviewer 3 Report

In this article, the authors develop an Artificial Neural Network, which, with spatial data obtained from IMUs, can discern whether a dog has unilateral hindlimb lameness. Also, the authors seek to develop a web portal with the capacity to manage dog lameness data to follow up with the dog over time.

The topic of the article is interesting, and the reader gets the required information. However, improvements and corrections are required.

1)At the end of the introduction the authors should present the structure of the article.

2)They should separate the Materials, Methods and Results section into two different sections. Material and Methods, and Results.

3)Μake a more detailed description of the parameters that you defined the Neural Networks.

4)Αll the figures in the article are PrintScreen, and their quality is too low.

5)Figures 9, 11, 12 and 13 should be as supplementary and the source code accessible.

6)The conclusions section should be better analyzed, and future directions of this work should be presented.

7)Extensive editing of English language and style required.
